# Amplifying and Fine-Tuning Rsm sRNAs Expression and Stability to Optimize the Survival of *Pseudomonas brassicacerum* in Nutrient-Poor Environments

**DOI:** 10.3390/microorganisms9020250

**Published:** 2021-01-26

**Authors:** David Lalaouna, Sylvain Fochesato, Mourad Harir, Philippe Ortet, Philippe Schmitt-Kopplin, Thierry Heulin, Wafa Achouak

**Affiliations:** 1Aix Marseille Univ, CEA, CNRS, BIAM, Lab Microbial Ecology of the Rhizosphere (LEMiRE), F-13108 Saint-Paul-Lez-Durance, France; d.lalaouna@ibmc-cnrs.unistra.fr (D.L.); sylvain.fochesato@cea.fr (S.F.); Philippe.ortet@cea.fr (P.O.); thierry.heulin@cea.fr (T.H.); 2Université de Strasbourg, CNRS, ARN UPR 9002, F-67000 Strasbourg, France; 3Research Unit Analytical BioGeoChemistry, Helmholtz Zentrum München, 85764 Neuherberg, Germany; mourad.harir@helmholtz-muenchen.de (M.H.); schmitt-kopplin@helmholtz-muenchen.de (P.S.-K.); 4Technical University of Munich, Maximus-von-Imhof-Forum 2, 85354 Freising, Germany

**Keywords:** Rsm sRNAs, *Pseudomonas brassicacearum*, sRNAs stability, nutritional stress, GacA-dependent

## Abstract

In the beneficial plant root-associated *Pseudomonas brassicacearum* strain NFM421, the GacS/GacA two-component system positively controls biofilm formation and the production of secondary metabolites through the synthesis of *rsmX*, *rsmY* and *rsmZ*. Here, we evidenced the genetic amplification of Rsm sRNAs by the discovery of a novel 110-nt long sRNA encoding gene, *rsmX-2*, generated by the duplication of *rsmX-1* (formerly *rsmX*). Like the others *rsm* genes, its overexpression overrides the *gacA* mutation. We explored the expression and the stability of *rsmX-1*, *rsmX-2*, *rsmY* and *rsmZ* encoding genes under rich or nutrient-poor conditions, and showed that their amount is fine-tuned at the transcriptional and more interestingly at the post-transcriptional level. Unlike *rsmY* and *rsmZ*, we noticed that the expression of *rsmX-1* and *rsmX-2* genes was exclusively GacA-dependent. The highest expression level and longest half-life for each sRNA were correlated with the highest ppGpp and cyclic-di-GMP levels and were recorded under nutrient-poor conditions. Together, these data support the view that the Rsm system in *P. brassicacearum* is likely linked to the stringent response, and seems to be required for bacterial adaptation to nutritional stress.

## 1. Introduction

The RNA-binding protein CsrA (carbon storage regulator) and its homolog RsmA (regulator of secondary metabolites) regulate a variety of genes at the post-transcriptional level. Their regulatory activity is modulated by regulatory small RNA (sRNA) antagonists of CsrA/RsmA. Functional homologs of the Csr/Rsm system have been discovered in many Gram-negative genera, including *Escherichia* [1], *Salmonella* [2], *Vibrio* [3], *Legionella* [4], *Yersinia* [5] and *Pseudomonas* [6,7,8], as well as in some Gram-positive bacteria such as the *Bacillus* genus [9].

The wide distribution of the Csr/Rsm regulatory system in Eubacteria, including animal/plant pathogenic species as well as plant growth-promoting rhizobacteria (PGPR), illustrates that this well-conserved system is involved in various essential functions [10,11]. Indeed, small RNAs positively controlled by either the BarA/UvrY or GacS/GacA two-component systems have been shown to regulate various functions by coordinating the expression of a set of target genes post-transcriptionally [12,13]

The number and size of sRNAs involved in this system are variable. In *Escherichia coli*, the protein repressor CsrA is regulated by two sRNAs, CsrB and CsrC. These sRNAs sequester CsrA by mimicking its recognition sequence [14,15]. Here, CsrB sRNA, a 360-nucleotide long RNA, works in concert with CsrC sRNA (250 nts). Two homologs were found in *Pseudomonas aeruginosa* and named RsmY (124 nt) and RsmZ (115 nt) [7]. A third Rsm sRNA, RsmX (119 nt), was discovered in *Pseudomonas protegens* (formerly *Pseudomonas fluorescens*) CHA0 [6] and predicted to be found in other species including *P. fluorescens* Pf-5, *P. fluorescens* Pf0-1, *Pseudomonas putida* KT2440 and *Pseudomonas syringae* pv. tomato DC3000. Surprisingly, up to five RsmX encoding genes have been described in *P. syringae* [8]. Moreover, experiments with artificial RsmY-like sRNAs have demonstrated that conservation of size and/or primary sequence is not essential for Rsm functions [16]. In fact, the regulatory mechanism is linked to appropriate exposure of mimicked CsrA/RsmA recognition motif.

In general, Csr and Rsm sRNAs seem to be redundant [7,17,18]. However, although all Rsm sRNAs are positively controlled by GacA, additional regulators have been identified, such as PsrA, which activates *rsmZ* in *P. protegens* CHA0 [19] and HptB, which exclusively controls *rsmY* in *P. aeruginosa* [20]. These additional regulators certainly enable a fine-tuning control by integrating various signals.

We have previously, shown that spontaneous mutations in either *gacS* or *gacA* in *P. brassicacearum* can alter many phenotypes typically associated with phytobeneficial traits, as well as root colonization and modification of the root architecture [18,21,22,23]. These traits include secretion of exoenzymes such as protease and lipase [24], production of antifungal metabolites (DAPG and cyanide) and *N*-acyl-homoserine lactones, the phytohormone auxin, biofilm formation, alginate production, chemotaxis, and type VI secretion [18]. In addition, overexpression of either *rsmX*, *rsmY*, or *rsmZ* fully suppressed the pleiotropic *gacA* or *gacS* mutations and restored the WT phenotypes in *P. brassicacearum* variants [18].

The primary goal of this study was to determine whether *P. brassicacearum* genome encodes other Rsm sRNAs and whether their overexpression could override the *gacA* or *gacS* mutation. The second part of our work aimed at evaluating the expression and stability of Rsm sRNAs under conditions of different nutrient levels. Finally, to establish the reliability of the results, the analysis of metabolites involved in stringency and biofilm formation was carried out.

## 2. Experimental Procedures

### 2.1. Bacterial Strains, Plasmids and Growth Conditions

The bacterial strains and plasmids used in this study are listed in Appendix A. *P. brassicacearum* NFM 421 and *gacA* mutant were grown as described in Lalaouna et al. [18]. *Escherichia coli* GM2163, TOP10 and S17-1 strains were grown in LB medium at 37 °C.

In order to test influences of carbon availability, bacterial strains were grown in TSB (BD), TSB/10 and TSB/20 media. For growth on plates, media were solidified with 15 g/L agar (Sigma, St. Louis, USA). Pseudomonas agar F (PAF) (Difco) was used to compare bacterial colony morphology of NFM421 WT, *gacS* and *gacA* mutants.

### 2.2. DNA Manipulation

Chromosomal DNA from *P. brassicacearum* NFM421 was prepared by phenol/chloroform extraction. Plasmid extraction using the “QIAprep Spin Miniprep Kit” (Qiagen, Hilden, Germany), and purification of DNA fragments from agarose gels with the “QIAquick Gel extraction Kit” (Qiagen, Hilden, Germany) were performed according to the manufacturer’s instructions.

### 2.3. RNA Manipulation

For RNA extractions, we used “RNAprotect Bacteria Reagent” and “RNeasy Mini Kit” (Qiagen, Hilden, Germany). RT-PCR assays were performed with the “Transcriptor First Strand cDNA Synthesis Kit” (Roche, Manheim, Germany). Gene-specific primers for real-time PCR were designed based on *P. brassicacearum* NFM421 *rsmX-1*, *rsmX-2*, *rsmY* and *rsmZ* sequences to obtain predicted PCR products of 200–250 bases (Appendix A). Amplifications were performed according to the real Q-PCR Light Cycler 480 SYBR Green I Master kit instructions for the Light Cycler 480 Real Time PCR System (Roche). Real-time PCR was performed in triplicate, and mRNA relative expression was normalized to the 16S reference gene.

### 2.4. 5′ RACE

The 5′-end of the *rsmX-2* transcript was mapped by RACE. Total RNA from *P. brassicacearum* strain NFM421 isolated in late stationary phase was used for a reverse transcription reaction with the Transcriptor First Strand cDNA Synthesis Kit (Roche, Manheim, Germany). The 5′-phosphorylated, 3′-end cordecypin-blocked oligonucleotide DT88 [25] was ligated to the single-strand cDNA with T4 RNA ligase (New England Biolabs, Ipswich, USA). The anchor-ligated cDNA was first amplified with primers DT89 (anchor-specific primer) and a *rsmX-2*-specific primer (*rsmX-2*-R1; Appendix A). Next, a nested PCR was performed using DT89 and a second *rsm*-specific internal primer (*rsmX-2*-R2). Finally, the PCR product was cloned into the cloning vector pCR^®^4Blunt-TOPO^®^ (Thermo Fisher Scientific, Waltham, USA). Three independent clones were sequenced using T7 primer (GATC Biotech, Konstanz, Germany).

### 2.5. Overexpression of sRNAs

To overexpress *rsmX-2*, we used the pME6032 plasmid [26]. The Shine-Dalgarno sequence was removed, as it is unsuitable for the expression of small RNAs. The *rsmX-2* gene was amplified from chromosomal DNA by PCR using primers *rsmX-2*-PstI and *rsmX-2*-KpnI (Appendix A), digested by PstI and KpnI, and inserted into PstI/KpnI-cut pME6032. The introduction of the PstI restriction site adds 5 nt and modifies 1 nt of the RsmX-2 sRNA at the 5′ end (5′-CTGCAGCCACTG… in place of 5′-TCCACTG). All mutations were verified by sequencing the inserts.

### 2.6. Construction of Transcriptional lacZ Fusions

To construct the *rsmX-2* + 16 transcriptional fusion, we amplified the promoter region and the first 16 nts of rsmX-2 sequence by PCR with primers listed in Appendix A. The PCR products were subsequently digested with specific restriction enzymes and cloned into pME6016 plasmids [27] and then sequenced. The pME6016-*rsmX-2*-*lacZ* plasmid was introduced into competent cells of NFM421 strain by electroporation ((2500 V for 5 ms using an Eppendorf Multiporator^®^ electroporator, Hambourg, Germany).

### 2.7. Northern Blot Analysis

RNA was extracted from overnight cultures of NFM421 WT strain and from ∆*gacA* mutant using “RNeasy Mini Kit” (Qiagen, Hambourg, Germany). The electrophoresis of total RNA was performed on a polyacrylamide gel (5% acrylamide 29:1, 8 M urea). RNAs were then transferred on a Hybond N+ nitrocellulose membrane (GE Healthcare Life Sciences, Uppsala, Swenden). RsmX-2 or 5S-specific digoxygenin-labelled probes were used (Appendix A). 5S rRNA was used as loading control. Luminescent detection was carried out as previously described [28].

### 2.8. Protease Activity Assays

In order to detect the extracellular protease activity, the bacteria are inoculated on TSB/10 solid medium containing 1% skim milk powder (TSA lec). The presence of a halo around the colony attests to protease activity.

### 2.9. β-Galactosidase Assays

Cultures containing a *rsmX-2* + 16-*lacZ* construct were grown overnight, diluted 1:200 into 8 mL of TSB with tetracycline (20 µg/mL), and grown over a period of 24 h. For the time-course of *rsmX-2* expression, 50 mL cultures were sampled at different times and assayed immediately. β-galactosidase activities were quantified as previously described [18]. Experiments were conducted in triplicates.

### 2.10. Biofilm Assays

As previously described [18], overnight cultures were diluted to OD_600nm_ = 0.05 in K10T-1 medium [29] and 1 mL aliquots were dispensed into glass tubes in triplicate. Following static incubation at 30 °C, the medium was removed, and tubes were washed gently with distilled water. Biofilm formation was visualized by crystal violet staining.

### 2.11. Stability of Rsm sRNAs

Wild-type cells were grown in TSB/10 or TSB media. To arrest RNA transcription, rifampicin was added at a final concentration of 200 µg/mL, either in early stationary phase (OD_600nm_ 1.5 in TSB and 0.5 in TSB/10) or in late stationary phase (OD_600nm_ 4.5 in TSB and 0.9 in TSB/10). At different time points (1, 2, 5, 10, 15, 20, 30, 45, 60, 90 and 120 min), 0.5 mL from each sample was mixed with 1 mL of RNA protect (Qiagen, Hilden, Germany), and total RNAs were extracted as described above. TURBO DNase treatment following reverse transcription, TaqMan real-time PCR analysis was used to determine the amount of each Rsm.

TaqMan PCR reactions were performed as described in the “LightCycler^®^ 480 Probes Master” (Roche) protocol, with double-labelled oligonucleotide probe (FAM and TAMRA dyes in 5′ and 3′ ends, respectively; Eurogentec). Reactions were run in a LightCycler^®^ 480 System. Threshold detection parameters were determined using the second derivative method. PCR efficiencies were determined by amplification of four 10-fold serial dilutions of all target sequences. Triplicated biological samples were quantified, and mean values were used to express Rsm sRNAs abundance relative to 16S RNA.

### 2.12. High Resolution Mass Spectrometry Analysis

WT cells were grown in TSB/20, TSB/10 or TSB media. Cells were harvested in late stationary phase and washed with ultrapure water. The lyophilized samples were extracted in methanol water (*v*:*v*) in an ultrasonic bath for 15 min. The pellets were centrifuged at 14,000 rpm for 5 min and the supernatants were analyzed in negative mode using 12 Tesla SOLARIX Fourier transform ion cyclotrom mass spectrometer (FT-ICR/MS) from Bruker Daltonics, Bremen, Germany. The injections were performed using a micro-liter pump at a liquid flow rate of 120 μL h^−1^. Nitrogen was used for both sheath gas as well as curtain gas. A source heater temperature of 200 °C was maintained to ensure rapid solvent evaporation in the ionized droplets. Data acquisition and handling were performed by using Data Analysis Software from Bruker (Bruker Daltonics, Bremen, Germany).

### 2.13. MassTRIX Metabolite Annotation

Samples were annotated to possible metabolites of *P. brassicacearum* using the KEGG metabolome database. The exact mass lists (asc files) were uploaded and compared with a 1.0 ppm accuracy window to the metabolome mass translator into pathways (MassTRIX) [30].

## 3. Results

### 3.1. Identification of a Fourth Rsm sRNA in P. brassicacearum

We identified a fourth Rsm sRNA by performing a BLAST search against the *P. brassicacearum* NFM421 genome and using the upstream activating sequence (UAS; TATAGCGAAACGCTTACA) recognized by GacA as query sequence. This revealed an additional UAS located upstream the gene encoding a putative sRNA (Figure 1A), which shares 75% similarity with *rsmX* gene, while the UAS sequence is almost identical (Figure 1B). We renamed RsmX to RsmX-1, and the novel Rsm sRNA was named RsmX-2. The secondary structure of RsmX-2 was predicted with RNAfold and visualized using vaRNA software [31]. At least three potentially mimicked recognition motifs of RsmA/E (i.e., ANGGA or AGGA motifs) [32], are exposed within a hairpin loop, suggesting that RsmX-2 could sequester these proteins (Figure 1C).

To determine, the relationship and evolutionary history, we performed a phylogenetic analysis of Rsm sRNAs genes. Phylogenetic tree Unweighted Pair Group Method with Arithmetic Mean (UPGMA) tree was constructed using MAFFT toolkit with a modified version of UPGMA. Rsm sRNAs exhibit a remarkably uniform distribution among *Pseudomonas* species. The UPGMA phylogenetic tree shows that *rsmX*, *rsmY* and *rsmZ* are sorted into three clades (Figure 2).

Typically, Pseudomonads contain single copies of RsmY and RsmZ, however, the copy number of RsmX is variable. *P. protegens* CHA0 possesses one copy of RsmX [6], while *P. syringae* contains up to five copies [8]. In this study we identified a fourth Rsm RNA in *P. brassicacearum* NFM421 found in tandem with *rsmX* with which it shares 75% of sequence identity. We also show a potential fourth Rsm sRNA in *P. fluorescens* F113 and *Pseudomonas stutzeri*, which are in tandem with *rsmX* and share 75% and 81% sequence identity respectively (Table 1).

### 3.2. rsmX-2 Expression Is Exclusively GacA-Dependent

The Northern blot analysis (Figure 3A) and 5′ end determination of the transcript by 5′RACE method confirmed that the novel 110-nt long Rsm sRNA is transcribed only in WT and not in the ∆*gacA* mutant. This was confirmed by the analysis of *rsmX-2* expression in wild-type and ∆*gacA* mutant cells during growth in 10-fold diluted TSB medium, by measuring β-galactosidase activity of the transcriptional *rsmX-2* + 16-*lacZ* fusion (Figure 3B).

Moreover, the overexpression of *rsmX-2* from a vector under the control of P*tac* promoter restored the wild-type phenotype in *gacA* mutant cells (Figure 4), indicating that the overall function of RsmX-2 is likely similar to that of RsmX-1, RsmY and RsmZ [18]. This is illustrated by the restoration of protease activity (Figure 4A), colony morphology (Figure 4B) and biofilm formation in a *gacA* mutant (Figure 4C). These findings indicate that RsmX-2 is part of the Gac-Rsm regulatory system.

### 3.3. Stringent Conditions Activate rsm sRNAs Genes Expression

The expression of the four *rsm* genes *rsmX-1*, *rsmX-2*, *rsmY* and *rsmZ* increases over time until a maximum is reached in stationary phase (24 h of growth), when nutrients become limited. We therefore wondered whether carbon deprivation could activate the *rsm* expression, as previously reported for the Csr system [33]. To assess the biological relevance of these observations, we monitored the expression level of the four *rsm* genes under different nutrient level conditions. Here, bacteria were grown in either undiluted TSB medium (considered as a rich medium), 10-fold diluted TSB (TSB/10), or 20-fold diluted TSB (TSB/20) (which was considered as a nutrient-poor medium in which *P. brassicacearum* growth may occur). Using transcriptional fusions for the promoter from each *rsm* gene, we found a greater increase in *rsm* expression when nutrient availability declines (Figure 5). Thus, even if growth decreases when nutrients are scarce, as indicated by OD_600 nm_ monitoring in tables beneath each graph of Figure 5, the expression of the four *rsm* genes highly increases. The greatest effect was observed for *rsmZ* (Figure 5). Nutrient starvation conditions influenced the expression levels of *rsm* genes and also certain targets of RsmA/E synthesis, as indicated by an enhancement of protease activity (Appendix A).

### 3.4. Stability of Rsm sRNA

If most sRNAs indeed regulate translational efficiency, their turnover in varying environments deserves a closer examination. For this, we examined the stability and steady state level of Rsm sRNAs by quantitative real-time reverse transcription polymerase chain reactions (qRT-PCR). We used two different media: TSB/10 and undiluted TSB medium. To determine the in vivo stability of Rsm sRNAs, exponentially phase and late stationary phase growing bacteria were treated with rifampicin to block any further initiation of transcription. Samples were taken at 1, 2, 5, 10, 15, 20, 30, 45, 60, 90 and 120 min and DNase-treated total RNA was used for qRT-PCR (Table 2, Appendix A). The half-life of each Rsm sRNA was calculated, revealing that these sRNAs are not equally stable. Overall, the half-lives of RsmY and RsmZ are much longer than those of RsmX-1 and RsmX-2 (Table 2, Appendix A).

The half-life of the four Rsm sRNAs increased during growth to reach their maxima during the stationary phase. For example, RsmX-1 and RsmX-2 stability increased ~8-fold in TSB/10 in the stationary phase compared to the exponential phase. RsmY and RsmZ were observed to be highly stable with a half-life of at least 60 min (Table 2, Appendix A).

Moreover, the choice of nutrients condition alters their stability; the four Rsm sRNAs were observed to be more stable in TSB/10 medium at stationary phase (e.g., half-life of RsmY and RsmZ is >60 min and ~50 min in TSB/10 and TSB, respectively). A similar effect was observed for RsmX-1 and RsmX-2, whereas, they both exhibited half-lives of 30–34 min in TSB/10 medium, which were reduced in TSB medium to 23 and 16 min, respectively.

In *P. brassicacearum*, we hypothesize a correlation between the number of GGA motifs and the stability of the four Rsm sRNAs transcripts conferred by RsmA/E. RsmZ, which showed the highest stability, possesses the highest number of GGA motifs (10 with 4–8 exposed); in contrast, RsmX-1 and RsmX-2 have the lowest number of GGA motifs (6 with 3–4 exposed) and were less stable. It should be noted that RsmY is moderately stable, although it only contains 7 GGA motifs (4–5 exposed) (Table 3).

### 3.5. Stringent Conditions and Sedentary Lifestyle Signalling Molecules

Given that the Rsm sRNAs expression and stability were increased under starvation conditions and that the bacterial stringent response, is mediated by the ppGpp, we evaluated the production of ppGpp under these culture conditions. Spectra were acquired with a time domain of 4 MW and a total number of 750 scans were accumulated (Figure 6A). As expected, the level of ppGpp is inversely proportional to the availability of nutrients (Figure 6B), while it evolves in the same way as the expression of Rsm sRNAs and in particular *rsmZ* [34].

Moreover, since Rsm sRNAs promote sedentary state and multicellularity through biofilm formation, a process also coordinated by the bacterial second messenger, cyclic diguanylate monophosphate (c-di-GMP), we also measured its cellular level. The same trend as for ppGpp and Rsm sRNAs was observed, suggesting a link between Gac-Rsm system and these two signal molecules (Figure 6C) as previously shown for other *Pseudomonas* species [35,36,37].

## 4. Discussion

### 4.1. P. brassicacearum Rsm sRNA Amplification by a Duplication of rsmX

The evolutionary relationships between the *rsmX* genes indicate the duplication events that occurred in different *Pseudomonas* species (Figure 2) with up to 5 copies in *P. syringae* was reported by Moll et al. [8], who correlated the number of Csr/Rsm sRNAs with the number of Csr/Rsm proteins. In *E. coli* and in most *Pseudomonas* species, there are two sRNAs for one or two regulatory proteins. However, three to five RsmA homologs are predicted in *P. syringae*, where up to seven Rsm sRNAs are found [8]. In *P. brassicacearum*, three RsmA homologs are found: RsmA, RsmE and a third putative RsmA-like protein. Recently, Sobrero and Valverde [11] performed a comparative genomics and evolutionary analysis of Rsm RNAs-binding proteins of the CsrA family in the genus *Pseudomonas* and suggested that the presence of redundant Rsm proteins that can replace or by-pass each other’s activities could help bacteria achieve greater plasticity via post-transcriptional regulation and better noise control in gene expression.

The amplification of regulatory RNAs is not exclusive to Rsm sRNAs. In *Vibrio cholerae*, four Qrr (quorum regulatory RNA), which are almost 80% identical in sequence and predicted to have similar secondary structures [3], are involved in virulence and biofilm formation [38]. Furthermore, five Qrr sRNAs are involved in the *Vibrio harveyi* quorum-sensing cascade [39]. In different bacterial species, two quite identical sRNAs that control iron homeostasis, as PrrF1 and PrrF2 in *Pseudomonas* [40] and, RyhB1 and RyB2 in *Salmonella* [41].

Gene duplication is an important feature in evolution because it provides raw material for adaptation to environmental challenges [42]. This phenomenon is also implicated in enabling gene amplification, and allowing cells to proliferate under growth-limiting conditions [43]. Any environmental condition favouring cells with more copies of a gene would permit them to outgrow the rest of the population in a short time; this includes duplications of rRNA genes that might confer a selective advantage under fast growth rate conditions [44]. Other cases of gene amplification are found in response to antibiotics [45].

Gene duplication-amplification is a frequent process in bacterial genomes, that often disappears after only a few generations of growth in the absence of selection pressure. However, gene copy (ies) may be preserved if they confer an adaptive evolution by increasing gene dosage in response to certain environmental constraints. We previously showed the key role of Rsm sRNAs in activating beneficial traits in plant-associated bacteria as well as virulence genes in phytopathogens [10,18].

### 4.2. RsmX-2 Is Part of Gac-Rsm System under Exclusive Control of GacA

Despite the similarity between the *rsmX-1* and *rsmX-2* sequences, the expression of *rsmX-2* differs from that of *rsmX-1*. The few mutations in the UAS (Figure 1B) of *rsmX-2* may explain the 8-fold decrease in the expression level. Our data confirm that RsmX-2 sRNA is a member of the Rsm sRNAs family. Indeed, the exclusive overexpression of *rsmX-2* in the *gacA* mutant suppresses the effect of *gacA* deletion, suggesting that the four Rsm sRNAs share a redundant function, which appears to be the sequestration of RsmA/E proteins.

In the absence of GacA in *P. aeruginosa* and *P. brassicacearum*, transcription of *rsmY* and *rsmZ* is still achieved but to a lesser degree, suggesting the involvement of additional regulatory pathways [7,18]. We previously demonstrated the activation of *rsmZ* expression, which depends on an enhancer sequence located in the *rpoS* coding region. This suggested the role of additional transcriptional factors as part of the complex network controlling the expression of *rsmZ*. We showed that the conserved palindromic UAS required for GacA-controlled sRNA genes in Gammaproteobacteria is essential but not sufficient for the full expression of the *rsmZ* gene in *P. brassicacearum* NFM 421 [46].

Finally, unlike the two novels recently discovered sRNAs, RsmV [47] and RsmW [48], in *P. aeruginosa*, which transcription is independent of the GacS-GacA, *rsmX-1* and *rsmX-2* expression is exclusively GacS-GacA-dependent.

### 4.3. Regulation of the Amount of RNA by Degradation

sRNAs often modulate mRNA targets stability, notably by recruitment of RNases [49]. This indicates that degradation is a regulatory process, by which bacteria adjust the translation of certain transcripts in response to changing environments. However, turnover of Csr/Rsm sRNAs has not been well-reported in the literature, with the exception of a study reported by Suzuki et al. [50]. These authors identified a regulatory protein (CsrD) that targets the global regulatory RNAs CsrB and CsrC for degradation by RNase E. Investigation of Rsm sRNAs degradation in *P. brassicacearum* indicates that the half-lives of the four Rsm sRNAs increase in the stationary phase and when nutrients are limited. The rate of turnover for Rsm sRNAs thus seems to be related to growth conditions and cell physiology. Furthermore, the four Rsm sRNAs are not equally stable, with RsmZ appearing to be extremely stable even during the exponential phase (Table 2).

Regulation of RNA amount by degradation is achieved by ribonucleases (RNases), whose activity is dependent on the sequence and/or the structural elements of the RNA molecule. In *Salmonella enterica* serovar Typhimurium, the endoribonuclease RNase III has been shown to regulate MicA (an sRNA involved in porin regulation) in a target-coupled way, whereas RNase E is responsible for the control of free MicA levels in the cell [51].

We hypothesize that the observed differences in Rsm sRNAs are mainly due to the sequence and structural elements of each Rsm sRNA. In *P. protegens* CHA0, the estimated half-lives of RsmY and RsmZ are >20 min in the wild-type and <10 min in the *rsmA rsmE* double mutant [52]. RsmA/E appears to stabilize Rsm sRNAs in vivo probably by protecting them from degradation by RNases. In *Erwinia carotovora* subsp. *carotovora*, RsmA also increases the half-life of the RsmB riboregulator [53]. Higher levels of RsmB RNA in the *rsmA^+^* strain than in the *rsmA^−^* strain were shown to be due to the increase in RsmB stability and not to an increase in transcription.

It has been suggested that RNase E and Hfq have similar (AU-rich) target sequences that should permit Hfq to protect sRNA from RNase E attack [54]. In *P. aeruginosa*, RsmY has been described as stabilized by Hfq, which sequesters RNase E cleavage sites [55], and thus enhances RsmA sequestration by RsmY [56]. In *P. brassicacearum*, an interaction of Hfq with RsmY and RsmZ may explain the extreme stability of these two sRNAs. Nevertheless, in *P. aeruginosa*, Hfq doesn’t affect RsmZ stability, whereas in *P. brassicacearum*, RsmZ exhibits an A-U rich loop, which could interact with Hfq.

### 4.4. Modulation of Rsm sRNAs Expression in Response to Nutrient-Poor Conditions

The stimuli that activate the Gac-Rsm system are still unknown [57]. In this study, we demonstrated that *rsm* genes are highly up-regulated under nutrient-poor conditions. Even if Rsm sRNAs are principally synthesized in the stationary phase, our results indicate that their expression is not only related to cellular density, but also and essentially to the physiological and metabolic state of cells. According to carbon source nature and availability, certain metabolites may accumulate within cells and induce specific responses. In *P. protegens* CHA0, pools of 2-oxoglutarate, succinate, and fumarate were shown to be positively correlated with the expression level of *rsm* genes [58]. Moreover, the expression of Rsm sRNAs is attenuated in the double mutant *relA spoT*, for which ppGpp synthesis has been completely abolished [34]. Our data show an activation of (p)ppGpp synthesis in response to nutrient starvation, correlating well with an increase of the four Rsm sRNAs in *P. brassicacearum* NFM421.

Another signaling molecule, the cyclic-di-GMP second messenger, which modulates the transition from the planktonic to the biofilm state [59], was shown to be positively regulated by GacS-GacA system in *P. aeruginosa* [35,37]. More recently, Liang et al. [36] showed that the QS system and polyketide antibiotic 2,4-DAPG production are regulated by c-di-GMP through RsmA and RsmE proteins in *P. fluorescens* 2P24. Our work also suggests a connection between the Gac-Rsm cascade and the c-di-GMP signaling pathway in phytobeneficial *Pseudomonas*.

Surprisingly, under circumstances that are not advantageous for *P. brassicacearum* proliferation, such as when nutrients are depleted, bacteria favor the activation of *rsm* expression and enhance their stability and increase (p)ppGpp and c-di-GMP intracellular levels. Consequently, production of certain secondary metabolites and biofilm formation might be enhanced, suggesting their importance in the adaptive response to environmental constraints.

Based on the present results and those of previous studies [34,36], we propose a model according to which ppGpp as well as c-di-GMP are directly or indirectly positively regulated by GacS/GacA system under nutrient limitations conditions (Figure 7).

## 5. Conclusions

The amplification of Rsm sRNAs and the preservation of several active copies indicate the importance of this post-transcriptional regulation, which allows the bacterium to increase its plasticity by rapidly adjusting its physiology in response to environmental constraints.

The existence of multiple factors devoted to *rsm* expression and degradation, illustrates that bacteria have evolved versatile mechanisms to control Rsm sRNAs levels. The higher expression level and the greater stability of the four Rsm sRNAs when nutrient conditions are limited underscore their relevance in the ecological niche of *P. brassicacearum,* a soil in which nutrients are poorly available. It is particularly interesting to note that this transcriptional activation of the four Rsm sRNAS occurs in concert with the increase in the intracellular level of the two signal molecules, (p)ppGpp and c-di-GMP under stringency conditions.

The differences observed in expression and stability between the four Rsm sRNAs illustrate that the subtleties of their respective roles in bacterial adaptation to challenging conditions remain far from being completely understood, as well as the regulation network involving (p)ppGpp and c-diGMP. This definitively deserves to be investigated in further detail.

Finally, although Rsm sRNA concentrations in the cell vary due to noise in gene expression and transcript stability, the overall combined concentrations of each Rsm sRNA are probably maintained in homeostatic balance.

## Figures and Tables

**Figure 1 microorganisms-09-00250-f001:**
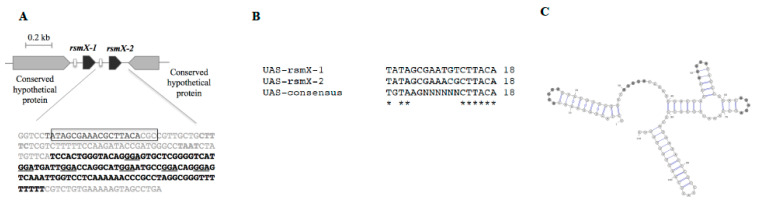
*rsmX*-2 identification and predicted structure. (**A**) Organization of the *rsmX-1* and *rsmX-2* genomic region of *P. brassicacearum*. The upstream activating sequence (UAS) of the GacA protein is represented within the box. GGA motifs involved in the sequestration of RsmA/E proteins are underlined. The transcriptional start has been determined by 5′RACE method. *rsmX-2* gene sequence is indicated in bold; (**B**) Alignment of *rsmX-1* and *rsmX-2* UAS sequence; (**C**) *rsmX-2* secondary structure predicted using RNAfold and visualised using vaRNA.

**Figure 2 microorganisms-09-00250-f002:**
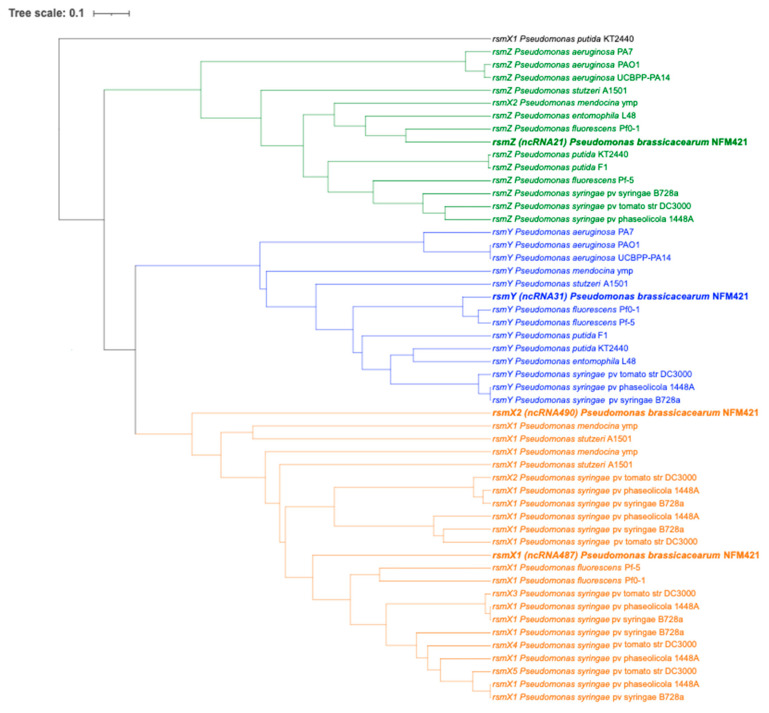
Phylogenetic analysis of *rsm* RNA sequences. *rsm* genes from *P. brassicacearum* NFM 421 are shown in bold.

**Figure 3 microorganisms-09-00250-f003:**
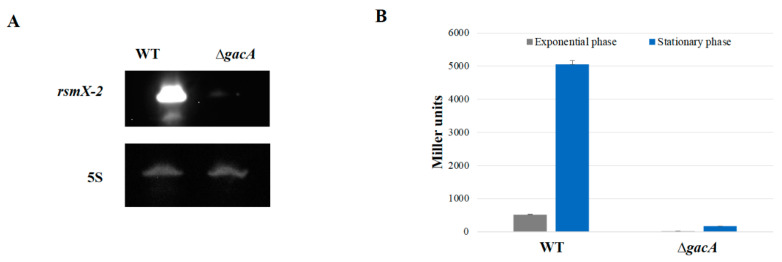
*rsmX-2* expression is GacA dependent. (**A**) Northern blot analysis of *rsmX-2* expression in WT and *gacA* backgrounds, 5S rRNA is used as loading control; (**B**) β-galactosidase measurements of a transcriptional *rsmX-2* + 16-*lacZ* fusion in the WT and ∆*gacA* strains.

**Figure 4 microorganisms-09-00250-f004:**
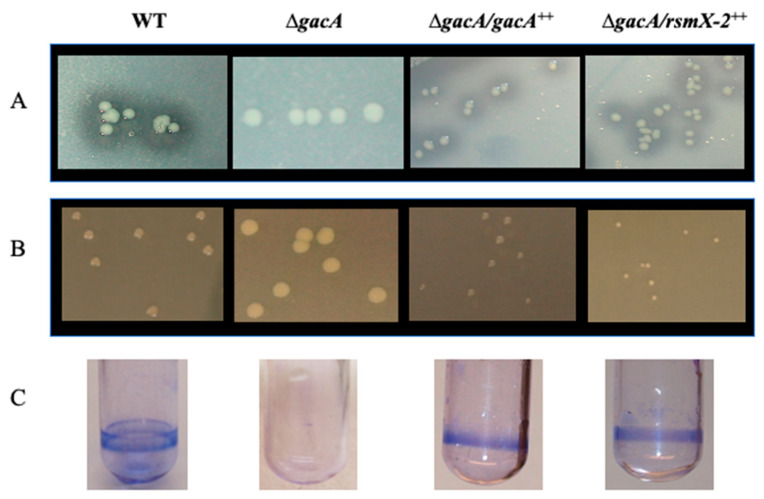
*rsmX*-2 restores GacA-dependent phenotypes. Restoration of wild-type phenotype by overexpression of *rsmX-2* from a plasmid under the control of P*tac* promoter in *gacA* cells as determined by (**A**) Protease activity on skimmed milk-containing medium, (**B**) Colony morphology on PAF medium and (**C**) Biofilm formation.

**Figure 5 microorganisms-09-00250-f005:**
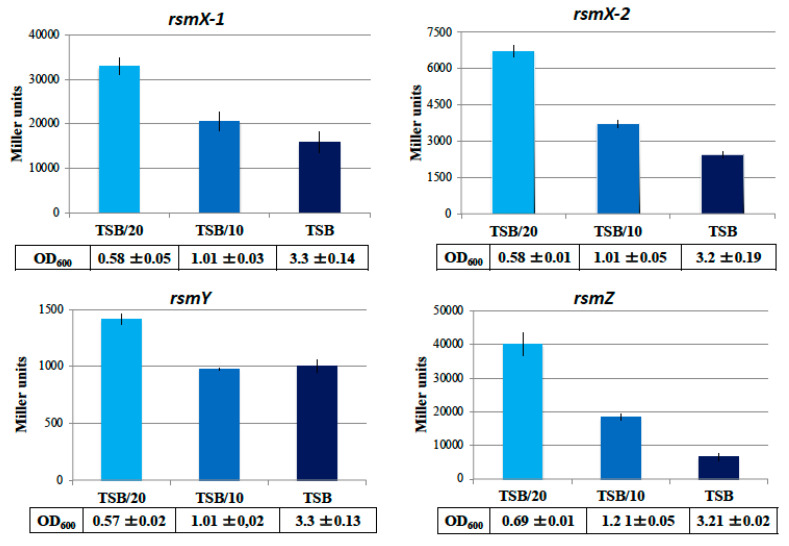
The expression level of *rsmX-1*, *rsmX-2*, *rsmY* and *rsmZ* depends on nutrient availability. Activity of a *lacZ* reporter fused to the promoter regions of *rsmX-1*, *rsmX-2*, *rsmY* and *rsmZ* were measured in the *P. brassicacearum* wild-type strain and are reported as Miller units of β-galactosidase activity (± standard deviation). The cultures were grown for 24 h at 30 °C before being tested and the tables below each graph indicate the OD_600_ of the cultures. Experiments were conducted in triplicates.

**Figure 6 microorganisms-09-00250-f006:**
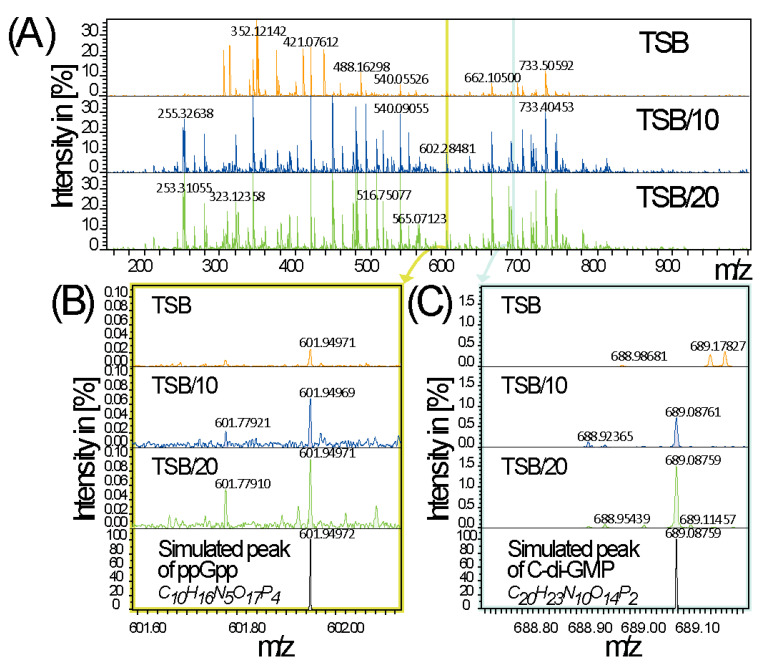
Signal molecules analysis. (**A**) Full ICR-FT/mass spectra of the wild type (WT) grown in TSB, tenfold dilutes TSB (TSB/10) and twenty-fold diluted TSB (TSB/20). (**B**,**C**) detection of (p)ppGpp and c-di-GMP compounds as annotated using metabolome database (MassTRIX: Suhre and Schmitt-Kopplin, 2008).

**Figure 7 microorganisms-09-00250-f007:**
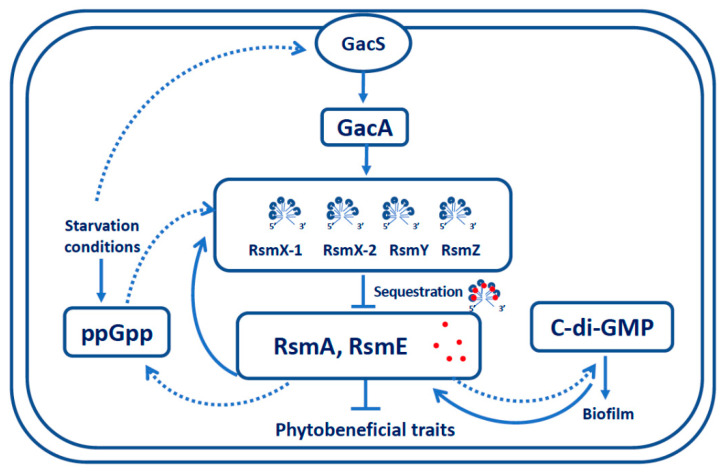
Model for gene regulation by ppGpp and c-di-GMP in the Gac-Rsm system of *P. brassicacearum* NFM421. This pathway is supported by evidence from this work and previous studies (Lalaouna et al., 2012, Takeuchi et al., 2012, Liang et al., 2020). ↓, positive effect; ┴, negative effect, dotted lines, indirect effects. The new conclusion of this study is that starvation conditions seem to activate the Gac-Rsm system concomitantly with the production of ppGpp and c-di-GMP.

**Table 1 microorganisms-09-00250-t001:** Tandem copy of rsmX gene in *Pseudomonas.*

Strains	Gene	Begin	End	Size (nt)	Sequence Identity (%)	Number of *rsmX-Like* Genes
*Pseudomonas brassicacearum* NFM421	*rsmX-1*	4839892	4840003	111	75	2
	*rsmX-2*	4839681	4839791	110		
*Pseudomonas fluorescens* F113	*rsmX-1 **	2113839	2113950	111	75	2
	*rsmX-2 **	2114052	2114160	108		
*Pseudomonas stutzeri* A150	*rsmX-1 **	356904	357013	109	81.5	2
	*rsmX-2 **	357143	357250	107		
*Pseudomonas syringae* pv. *phaseolicola* 1448A	*rsmX-4 ***	160448	160336	112	85	5
	*rsmX-3 ***	160740	160627	113		
*Pseudomonas syringae* pv. *syringae* B728a	*rsmX-3 ***	5867169	5867282	113	84	5
	*rsmX-4 ***	5867461	5867572	111		
*Pseudomonas syringae* pv. *tomato* str. DC3000	*rsmX-3*	6144830	6144943	113	87	5
	*rsmX-4*	6145122	6145235	113		

* in comparison to *P. brassicacearum* NFM421. ** in comparison to *P. syringae pv.* tomato str. DC3000.

**Table 2 microorganisms-09-00250-t002:** Stability of Rsm sRNAs. Half-lives of *P. brassicacearum* NFM421 RsmX-1, RsmX-2, RsmY and RsmZ. Experiments were conducted in triplicates.

Medium	Phase		Half-Lives in WT (in Min)			
		OD_600nm_	RsmX-1	RsmX-2	RsmY	RsmZ
TSB	Expo	1.5	12 ± 1.1	5 ± 0.6	8 ± 0.2	37 ± 2.6
	Stat	4.5	23 ± 5.4	16 ± 1.5	45 ± 5.2	53 ± 0.9
TSB/10	Expo	0.5	7 ± 1.5	5 ± 1.6	8 ± 1.7	26 ± 3.6
	Stat	0.9	37 ± 4.4	40 ± 8.6	≥60	≥90

**Table 3 microorganisms-09-00250-t003:** Rsm sRNAs stability is correlated to the number of GGA motifs. Relationships between the number of GGA motifs and Rsm sRNAs half-lives of *P. brassicacearum* NFM421.

			Half-Lives (in Min)	
Genes	GGA Motifs	GGA Motifs Exposed *	TSB/10 24 h	TSB 24 h
RsmX-1	6	2–4	37 ± 4.4	23 ± 5.4
RsmX-2	6	3	40 ± 8.6	16 ± 1.5
RsmY	7	5–4	≥60	45 ± 5.2
RsmZ	10	8–7	≥90	53 ± 0.9

* predicted with Mfold and RNAfold softwares.

## Data Availability

All data in this article is openly available without any restrictions.

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
