# Peer review of "Amplifying and Fine-Tuning Rsm sRNAs Expression and Stability to Optimize the Survival of Pseudomonas brassicacerum in Nutrient-Poor Environments"

_microorganisms, 2021, doi:10.3390/microorganisms9020250_

Round 1
Reviewer 1 Report
The authors submitted a new version of the manuscript that I previously reviewed. The manuscript has been improved in quality and all my requests were properly addressed.
Author Response
Dear reviewer,
Thank you.
Best regards
Wafa Achouak
Reviewer 2 Report
This study announced a novel sRNA encoding gene rsmX-2 in P. brassicacearum NFM421 (a fourth rsm gene); its expression was exclusively GacA-dependent and overexpression overrides the gacA mutation; nutrient-poor condition activated the expression of the four rsm genes, which was correlated with the accumulation of ppGpp and cyclic-di-GMP. The manuscript expands our knowledge with regard to the bacterial adaptation mechanism to nutritional stress, and this version has been well improved according to editor and reviewers' comments. I suggest acceptance after finishing a few revisions as listed below:
Page 2, Line 6-8 & 19, "Pseudomonas" can be abbreviated as "P. ".
Page 2, Line 15, "CHAO" should be "CHA0".
Page 2, Line 35, Table S2 should be S1, while Table S1 in line 52 should be S2.
Page 3, Line 4, should be "2.4. 5' RACE".
Page 5, Figure 2, the species and genus name in the tree should be italic.
Page 6, Table 1, please add "Strain" and "Gene" as the header for the left two columns; why the gene name in A150 is missing? For 1448A, B728a and DC3000, why only one or two rsmX gene are shown while there should be five copies?
Page 6, Figure 3A, "rsmx-2" should be "rsmX-2".
Page 8, Figure 5, please mark the statistical differences among treatments.
Page 8, Table 2, please add "Medium" and "Phase" as the header for the left two columns.
Page 8, Line 26, "media" or "nutrients condition"?
Page 9, Line 14, should be 3.5.
Page 11, Line 40, "CHAO" should be "CHA0".
The quality of all Figures should be further improved.
Author Response
Dear Reviewer,
We are grateful for your help in improving our manuscript. We apologise for any omissions, errors and imperfections. Please find below the answers to your comments. Changes have been made to the revised manuscript.
Page 2, Line 6-8 & 19, "Pseudomonas" can be abbreviated as "P. ".
We are sorry for these mistakes that we fixed.
Page 2, Line 15, "CHAO" should be "CHA0".
Mistakes fixed.
Page 2, Line 35, Table S2 should be S1, while Table S1 in line 52 should be S2.
Thank you for your comment. It was an oversight during the reorganization of the manuscript.
Mistakes fixed. There is only one supplementary Table in the revised manuscript.
Page 3, Line 4, should be "2.4. 5' RACE".
"2.4. 5' RACE" is correct
Page 5, Figure 2, the species and genus name in the tree should be italic.
You are right, genus, species and even genes names should be italic. These mistakes have been fixed.
Page 6, Table 1, please add "Strain" and "Gene" as the header for the left two columns; why the gene name in A150 is missing? For 1448A, B728a and DC3000, why only one or two rsmX gene are shown while there should be five copies?
All mistakes have been fixed.
For 1448A, B728a and DC3000, only copies of rsmX which are in tandem are shown in the table.
Page 6, Figure 3A, "rsmx-2" should be "rsmX-2".
This mistake has been fixed.
Page 8, Figure 5, please mark the statistical differences among treatments.
The experiments were performed in triplicates and the standard deviations were calculated. The absence of overlapping error bars indicates significant differences.
Page 8, Table 2, please add "Medium" and "Phase" as the header for the left two columns.
This mistake has been fixed.
Page 8, Line 26, "media" or "nutrients condition"?
We replaced media by nutrients condition
Page 9, Line 14, should be 3.5.
This mistake has been fixed.
Page 11, Line 40, "CHAO" should be "CHA0".
This mistake has been fixed.
The quality of all Figures should be further improved.
The quality of figures has been improved. We replaced some figures such as Figure 6.
Reviewer 3 Report
In this manuscript, Lalouna et al. identified a second copy of the rsmX gene, rsmX-2, which encodes a small RNA. The authors demonstrate that this new sRNA is duplicated from rsmX and that all rsm genes are induced under nutrient starvation conditions. The authors show that the rsmX genes are under direct control of the GacA regulator and hypothesize that the Gac-Rsm pathway are part of a regulatory network necessary for the stringent response. The results shown by the author support their hypothesis and conclusions. Only, a few specific comments are listed below, which should be rectified:
Comments
- Page 5 Line 2: Typo. Please correct ‘nd’ to ‘and’
- Page 6 Line 21-22: ‘Moreover….mutant cells (Figure 3).’ Figure 3 does not show any overexpression experiments for rsmX-2 in the gacA Please show appropriate controls for this experiment.
- In page 7 lines 13-14 (We therefore….Csr system [33]), the authors mention that they wanted to determine the effect of carbon starvation on expression of rsm However, the results may be incorrectly interpreted because of the media conditions (Page 7 Line 16-17: undiluted and diluted TSB medium). TSB is a rich undefined (peptides, amino acids, glucose, phosphates) medium and diluting it would dilute all carbon, nitrogen and phosphate nutrients. As it stands, the only conclusion from Figure 5 is that general nutrient starvation activates rsm promoters. A more appropriate way to determine effect of carbon starvation would be by using a defined medium such as MOPS or M9 and then observing rsm promoter activity. First, strains should be grown in the complete medium to mid-log phase. Then they should be inoculated into the defined medium excluding the carbon source and then promoter activity should be determined. The methods description (page 2, line 38) and objectives should be appropriately modified.
- Since, amino acids can serve as both carbon and nitrogen sources, it would be very interesting to see how starvation by either limiting sugar sources or amino acids could influence rsm genes expression. However, these experiments are not necessary within the context of this study.
Author Response
Dear Reviewer,
We are grateful for your help in improving our manuscript. We are sorry for any omissions, errors and imperfections. You will find in the attached file the answers to your comments. Changes have been made to the revised manuscript.
Comments
- Page 5 Line 2: Typo. Please correct ‘nd’ to ‘and’
This mistake has been fixed.
- Page 6 Line 21-22: ‘Moreover….mutant cells (Figure 3).’ Figure 3 does not show any overexpression experiments for rsmX-2 in the gacA Please show appropriate controls for this experiment.
We are sorry it concerns Figure 4. This mistake has been fixed.
- In page 7 lines 13-14 (We therefore….Csr system [33]), the authors mention that they wanted to determine the effect of carbon starvation on expression of rsm However, the results may be incorrectly interpreted because of the media conditions (Page 7 Line 16-17: undiluted and diluted TSB medium). TSB is a rich undefined (peptides, amino acids, glucose, phosphates) medium and diluting it would dilute all carbon, nitrogen and phosphate nutrients. As it stands, the only conclusion from Figure 5 is that general nutrient starvation activates rsm promoters. A more appropriate way to determine effect of carbon starvation would be by using a defined medium such as MOPS or M9 and then observing rsm promoter activity. First, strains should be grown in the complete medium to mid-log phase. Then they should be inoculated into the defined medium excluding the carbon source and then promoter activity should be determined. The methods description (page 2, line 38) and objectives should be appropriately modified.
The medium TSB/20 contains 20-fold less carbon and other nutrient compounds found in the standard TSB medium which corresponded to 0.85g of Pancreatic Digest Casein and 0.25g Papaic digest of Soybean, 0.25g of NaCl and 0.12g of Dipotassium phosphate per liter. As indicated by OD measurements in Figure 5, OD600 varied between 0.5-0.6 in TSB/20 to up to 3 in standard TSB medium. Pseudomonas brassicacearum needs a carbon source to grow in M9 medium.
|
OD600 |
0.58 ±0.05 |
1.01 ±0.03 |
3.3 ±0.14 |
The cultures were grown for 24 hours at 30°C before being tested and the tables below each graph indicate the OD600 of the cultures.
Moreover, succinate was barely detected in cells grown in TSB/20 compared to standard medium (TSB) (unpublished data). We observed the same trend with fumarate and oxoglutarate.
- Since, amino acids can serve as both carbon and nitrogen sources, it would be very interesting to see how starvation by either limiting sugar sources or amino acids could influence rsm genes expression. However, these experiments are not necessary within the context of this study.
You are right. These experiments are relevant.

This manuscript is a resubmission of an earlier submission. The following is a list of the peer review reports and author responses from that submission.
Round 1
Reviewer 1 Report
The manuscript entitled “Amplifying and fine-tuning Rsm sRNAs expression and stability to optimize survival in nutrient-poor environments” authored by Lalaoma and colleagues, deals with the investigation of the expression and the stability of four Rsm encoding genes under rich or nutrient-starved conditions. The first problem I've found in the reviewed paper concerns the lack of use of the correct article format. MDPI journals have a specific template, not followed in organizing this article. I strongly suggest that the authors modify the article entirely according to the journal guidelines.
Introduction section: the introduction is well written. However, in the final part of this section, the description of the results obtained is clearly reported. This part should be removed from the introduction as it does not concern the current state of the art, but it is a finding made by the authors. Furthermore, the aims of the work are not well expressed.
Figures: the figures have a very low quality, and several panels make a single picture. This format makes impossible for the reader to follow a logical line. In addition, some panels are too small, resulting in serious difficulties of interpretation. Since the journal's guidelines do not impose a maximum number of figures, I suggest to divide the panels into separate figures, or at most composed of two/three panels maximum. Whenever possible, the authors should use differently coloured graphs for the different quantifications. Finally, panel letters should be reported in UPPERCASE e in bold for each panel.
Tables: some data reported in the tables make no sense. For example, in Table 1, 15 ± 0 is not possible. Moreover, the data should be reported following the scientific system: at least three significant numbers should be reported for each number (15.2 ± 0.01); decimal numbers should be separated by "dots" and not by "commas". In Table 2, standard deviations should be reported. Some table are also reported as figures. This is not correct.
Materials and Methods: this section is poorly described. M&M is a fundametal section, important for other researchers who want to reproduce the experiments. Please implement this section with additional information.
Results and Discussion: the paper contains really interesting results and discussion, but not well presented. I strongly suggest to the authors to check carefully the manuscript, organize it according to the MDPI guidelines, correct the different typing mistakes present in the text.
Reviewer 2 Report
This manuscript written by D. Lalaouna et al. discovered a fourth rsm gene, rsmX-2, in Pseudomonas brassicacearum strain NFM421, found that the expression of rsmX-1 and rsmX-2 is exclusively GacA-dependent, and all the four Rsm sRNAs (rsmX-1, rsmX-2, rsmY and rsmZ) had highest expression under nutrient-starved culture condition. This is a very interesting finding that makes it possible to increase the survival ability of Pseudomonas brassicacearum in nutrient-poor environments.
I feel there is a bit thing missing in the title, probably the Pseudomonas brassicacearum?
The tense in the introduction part needs an intensive check.
In the result section, there are some sentences that belong to the discussion section in my eye.
The quality of the figures can be improved, the resolution is not high enough for readers to have a clear view. Figure 1a,1b,1c,1d,1e,1f can be better layout designed, especially Figure1f with “A”,” B”,” C”.
Figure 2 do not have panel names, I would suggest integrating four panels into one figure, in this way, readers can see more differences among rsmX-1, rsmX-1, rsmY and rsmZ under different nutrient availability condition. What’s the table beneath each figure for? I am a bit confused about this. I would suggest using another statistical method.
The top borders and bottom borders of all the tables are missing, they look a bit weird in this way. Table 1 should be moved close to the text which mentioned this table.
There are tables and figures showing in the discussion section, please move them to the result section.
In the material and method section, TSB is a nutrient-rich medium for culturing microorganisms. I have a consideration that whether TSB/20, TSB/10 are really nutrient-poor? Did authors check the nutrient content and concentration of TSB/20 and TSB/10? Can you create a "real" nutrient-poor condition to culture the bacterium and its mutants?
Keep nutrient-poor, nutrient-starved et al these terms consistent.
I cannot really understand the sentence in lines 6-8 on page 4, please rephrase it.
There are some minor comments in the attached pdf.
